# Microstructure and Fatigue Damage of 316L Stainless Steel Manufactured by Selective Laser Melting (SLM)

**DOI:** 10.3390/ma14247544

**Published:** 2021-12-08

**Authors:** Zhentao Wang, Shanglei Yang, Yubao Huang, Cong Fan, Zeng Peng, Zihao Gao

**Affiliations:** 1School of Materials Engineering, Shanghai University of Engineering Science, Shanghai 201600, China; wangzhentao23@163.com (Z.W.); huangyb16@163.com (Y.H.); fc19960802@163.com (C.F.); pengzeng98@163.com (Z.P.); gzh_960303@163.com (Z.G.); 2Shanghai Research and Development Center for Key Technologies of Ultra-Intense Laser Processing, Shanghai University of Engineering Science, Shanghai 201600, China

**Keywords:** 316L stainless steel, selective laser melting, microstructure, fatigue damage

## Abstract

In this paper, 316L stainless steel powder was processed and formed by selective laser melting (SLM). The microstructure of the sample was studied using an optical microscope, and the fatigue failure of the sample and the characteristics of crack initiation and propagation were analyzed, providing a research basis for the application of SLM-316L. Due to the influence of microstructure and SLM process defects, the fatigue cracks of SLM-316L mainly emerged due to defects such as lack of fusion and pores, while the cracks of rolled 316L initiated at the inclusions near the surface of the specimen. After fatigue microcrack initiation of the SLM-316L specimen, due to the existence of shear stress and tear stress, the crack tip was passivated and Z-shaped propagation was formed. The existence of internal defects in SLM-316L made the microcrack initiation random and diverse. At the same time, the existence of defects affected the crack propagation in the form of bending, bifurcation and bridge, which made the main crack propagation deviate from the maximum load direction.

## 1. Introduction

Selective laser melting (SLM) technology is a kind of additive manufacturing technology with great development potential. Based on the discrete-stack principle, this technology uses the laser beam as a heat source under the control of a galvanometer, and is guided by computer-aided design data to melt the selective area of metal powder layer by layer, so that the metal powder accumulates into a solid [1,2]. Recent studies have shown that the SLM process exhibits good mechanical properties due to its rapid cooling and unique multi-scale (from nano to macro scale) heterogeneous structure [3,4,5]. SLM is not only a useful supplement to traditional processing methods such as casting, forging, welding and machining, but also a new manufacturing mode of metal parts. SLM has broad prospects for development in the biomedical, aerospace and energy industries [6,7].

316L stainless steel is a kind of ultra-low carbon austenitic stainless steel, which is widely used in the aerospace and automobile industries due to its good comprehensive mechanical properties [8,9]. In recent years, the precise fabrication of complex parts for the aerospace industry using 316L stainless steel by SLM has become a research hotspot [10]. Compared to traditional molding technology, SLM has the advantage of not requiring development of a mold and having a short manufacturing cycle. Despite these irreplaceable advantages, the complex thermal effect and the typical rapid melting and cooling involved [3,11] means that SLM austenitic stainless steel has inherent defects, such as internal defects (especially unmelted hole defects), surface defects and residual stress, which strongly affect the fatigue performance of the processed materials, thereby reducing the reliability of engineering applications [12,13]. At present, many scholars have conducted a lot of research on the powder characteristics, optimization of different process parameters, density and mechanical properties of 316L stainless steel [14,15,16]. However, there are few reports investigating the fatigue failure mechanism of SLM-316L, yet fatigue failure is one of the most common failure forms of structural components under alternating loads [17,18,19,20]. The research on fatigue failure of SLM-316L is of great significance to the development of SLM technology. Riemer et al. used the fatigue behavior of 316L manufactured by SLM to check the crack initiation and crack growth behavior [21]. The results obtained clearly show that 316L is a promising candidate for cyclically loaded parts manufactured by SLM. Primarily attributed to the high ductility that directly follows the SLM process, the 316L stainless steel shows fatigue properties similar to that of a conventionally processed material in its as-built model. Sarkar et al. studied the fatigue properties and fracture types of SLM stainless steel under different average stress modes [22]. The fatigue properties and fracture characteristics of the samples under different average stress modes were significantly different, were mainly manifested as the average stress in tensile mode was not conducive to the fatigue performance, while the average stress in compression mode could improve the fatigue performance. Pegues et al. studied the fatigue crack initiation characteristics and the mechanism of AM-304L by the in situ microstructure observation of an interrupted fatigue test [23], and considered that the microstructure refinement and the decrease in high angle grain boundary density (HAGBs) improved the fatigue properties of AM-304L. Although relevant studies have reported the fracture process of static tensile failure and the corresponding potential failure mechanism, there are few studies on the fatigue failure of SLM-316L, which is insufficient for such an important failure. In order to understand the fatigue failure mechanism of SLM-316L, a series of fatigue tests were carried out on SLM-316L and rolled 316L by using a HB250 electro-hydraulic servo dynamic fatigue testing machine. The fatigue fracture surfaces of the two were observed by scanning electron microscopy (SEM) and optical microscopy (OM), and then the fatigue failure mechanism and crack propagation of SLM-316L were analyzed. These results provide a new understanding for us to better understand the fatigue failure of SLM-316L stainless steel.

## 2. Test Materials and Methods

The composition of 316L powder and traditional 316L stain-less steel is shown in Table 1. Before laser additive manufacturing, the powder was heated in a drying furnace to remove the moisture on the surface of the metal powder and increase the fluidity in the powder to avoid defects such as pores inside the additive manufacturing sample.

The forming equipment used in this study was an HBD-100SLM, produced by Hanbon Technology Co., Ltd. (Shanghai, China), as shown in Figure 1d. The schematic diagram of the 316L laser selective forming process and the scanning strategy is shown in Figure 1a–c. The process parameters used in this study are shown in Table 2. Before printing, the oxygen in the molding chamber was reduced to less than 1000 ppm, 316L stainless steel was used as the substrate, and the shielding gas was argon. In the printing process, a rotation of 67°, layer by layer, was performed using the scanning strategy shown in Figure 1c. The forming samples were prepared according to the two-dimensional data of the software imported before the test and the process parameters in Table 2. After preparing SLM-316L sample, the sample was cut according to the size diagram in Figure 1e.

After cutting, the samples were sanded by sandpapers with different levels of roughness until the surface scratches were shallow. Next, the samples were polished until no obvious scratches were observed under the microscope. Once polished, the samples were etched with corrosives (HCl:HNO_3_ = 3:1) for about 10 s.

According to GB/T 26077-2010 standard, and in a room temperature (25 ± 3 °C) environment with relative humidity (40–60%), the rolling 316L and SLM-316L were subjected to fatigue tests using the symmetric tension-tension cycle of sinusoidal loading waveforms until the sample reached 10^6^ cycles or fracture. The stress ratio (R = σmin/σmax) and load frequency were 0.1 and 15 Hz, respectively. All fatigue samples were cyclically loaded at different stress ranges (∆σ = σmax−σmin). The ultimate fatigue strength (Nf ≥ 10^6^) of rolled 316L and SLM-316L was analyzed by an SN curve. All fatigue tests were carried out on a Zwick/Roell Amsler HB250 hydraulic servo material testing machine, and the fatigue fracture was observed and analyzed by an S-3400N scanning electron microscope (SEM).

## 3. Results and Discussion

### 3.1. Microstructure and Defect Analysis

The sample was based on the XOY surface as the bottom surface and the Z axis as the construction direction, as shown in Figure 1c, Figure 2a,b, respectively, which illustrates the metallographic microstructure of samples in the vertical and parallel directions of construction. Once the additive manufacturing commenced, the laser was directed in a predetermined path, melting the metal powder to form a molten micro-molten pool, where the molten metals were bonded to each other in order to achieve the bonding of atoms. Figure 2a shows the microstructure of the sample perpendicular to the additive manufacturing direction, and the traces formed by the laser scanning powder can be seen. Figure 2b shows the microstructure of the sample parallel to the additive manufacturing direction. Visibly, a lot of fish-scale-shaped channels are stacked together. As the heat source of nearly Gaussian heat, the laser beam had a higher energy density in the center and a lower energy density in the edge. During the process of melting, the penetration of powder scanned by the center of laser beam was greater than that scanned by the edge of laser beam. After a lot of pre-laid powders were scanned one-dimensionally, two-dimensionally and then three-dimensionally by laser beam, a lot of overlapping fish scale interfaces were formed. The liquid molten pool nucleated directly on the contact surface between the molten metal and the semi molten grains and grew epitaxially. In addition, at the bottom of the micro molten pool, the temperature gradient was high, and the nucleation grains grew along the direction perpendicular to the temperature gradient, forming a series of long columnar crystals that grew through multiple powder layers [24,25].

Figure 2a,b show there were some defects such as pores (in yellow circles) and unfused porosity (in yellow boxes), and the gaps appeared where the melting channels made contact. Weingarten et al. studied the pore composition of laser selective melting samples. Their research found that the H_2_O attached to the powder surface decomposed and saturated in the molten metal to form hydrogen when the laser interacted with the metal powder. In the subsequent solidification process of the metal, the hydrogen overflowed to form pores. Before the metal powder was melted, there were some voids between the powders. During the solidification of molten metal, the gas in the voids will also produce pores without overflow. Figure 2c shows the microstructure of rolled 316L. Many rectangular grains can be seen, and the grain size distribution is uniform. The formation of splash and unfused porosity is shown in Figure 3. In the process of laser selective additive manufacturing, laser irradiation melts the metal powder to form a liquid molten pool. With the accumulation of heat, and after reaching the boiling point of the molten pool, the metal vapor is formed. Under the action of recoil pressure, the molten pool splashes. The existence of spatter can be seen in Figure 3b. In addition, as a near Gaussian heat source, the laser has high central energy density and small edges. During the laser scanning process, the molten pool is unstable, which can easily to lead to incomplete fusion at the overlap between the weld bead and the weld bead [26]. The presence of non-fusion can be seen in Figure 3c, and there is still unmelted metal powder inside the non-fusion in Figure 3d.

### 3.2. Fatigue Test Analysis

During the course of this study, a laboratory hydraulic servo low-frequency fatigue testing machine was used to apply a load to simulate the actual service process of the sample. The national standard GB/T 3075-2008 “Metal Material Fatigue Test Axial Force Control Method” was followed, using the lifting method to perform the fatigue test with 10^6^ cycles for rolling 316L and laser selective additive manufacturing 316L. Fatigue data results are shown in Table 3. According to the data in Table 3, the fatigue limits of rolling 316L and SLM-316L specimens were 370 MPa and 200 Mpa, respectively, under the condition that no fracture occurred after 10^6^ times. Without breaking after 10^6^ times, the stress level of SLM-316L only reached 54% of that of rolled 316L. As the stress level increased, the fatigue life of rolled 316L decreased rapidly. The fatigue life of laser-selected area additive manufacturing 316L decreased slowly with the increase in stress level. Sometimes with the increase in stress level, life expectancy will increase. Figure 4a–c show the fracture morphologies of rolled 316 B5, and Figure 4a shows the macroscopic morphology of sample B5. Visibly, there are fewer internal defects, an obvious plastic deformation in the fracture, and obvious inclusion defects in the lower left corner of the fracture. The inclusion defects are located on the surface of the sample and spread radially around. Figure 4b shows an enlarged view of the fatigue source area of sample B5, which is also a further enlargement of Figure 4a. Noticeably, fatigue cracks originate from inclusion defects on the surface of the sample. After the crack is formed, it starts from the defect and spreads radially around. Defects under cyclic loading will cause the stress concentration to be higher than the surroundings, resulting in microcracks in the sample without a large external load. Figure 4c is an enlarged view of the defect of the fracture surface of sample B5, which is also a further enlargement of Figure 4b. One can see the existence of the tear ridge (inside the yellow ellipse), which is left when the cracks in the expansion process meet once they have expanded in all directions.

Figure 4d–f show the fracture morphology of SLM-316L sample S6. Figure 4d shows the macro fracture morphology of sample S6. The figure illustrates that there are defects such as sintered powder (in the yellow frame) and unfused pores (in the yellow circle) inside the sample. Figure 4e–h are an enlarged view of the fracture morphology of sample S6, which is also a further enlargement of Figure 4d. As shown in Figure 4e, you can see the tearing ridges produced by the cracks on the left that meet the defects caused by unfusion. The cracks will expand further after encountering the defects during the propagation process. Furthermore, as shown in Figure 4f, around the sintered powder, the cracks generated will expand to the surroundings. As shown in Figure 4g, cracks are generated on the surface of the sample, and then expand forward, and tear ridges will also be generated when intersecting with other cracks. This differs from rolled 316L in that, depending on the type of cracks, microcracks may occur on the surface of defects such as sintered powder, unfused, and pores. According to the location of the crack, the fatigue source of laser-selected area additive manufacturing 316L not only exists on the surface of the sample, but also exists inside the sample, and may also exist near the surface. When cyclic loading, the stress concentration is prone to occur near the defect, especially on the surface of some sharp notches, as the stress concentration factor is much higher than the surroundings. There are microcracks near defects such as sintered powder, LOF, pores, etc., indicating that these process defects promote the formation and propagation of cracks during the actual loading process of the sample. The materials used this time are added with the same parameters. There are defects in samples S1–S7, however some samples have many defects and some samples have few defects. As a result, the fatigue life of laser selective additive manufacturing of 316L is not obvious, or the fatigue life is increased by the increase in stress level. If other parameters of SLM-316L are selected, the internal defects of 316L may be reduced and the fatigue performance may be better, however the defect cannot be completely eliminated. Defects such as sintered powder, LOF, and pores are common in laser additive manufacturing. These defects can be divided into two categories according to the causes. The first type is pores, which are caused by unstable small holes in the sidewalls that are prone to local severe necking and collapse during laser additive manufacturing. The necking and collapse of the liquid metal on the sidewall of the small hole wraps the gas in the small hole to form pores and cavities. The second category is non-fusion defects. There are two reasons for the occurrence of non-fusion defects: The first aspect is that the instability of laser additive manufacturing causes the hole wall to collapse and the powder particles in the bubbles of the liquid molten pool are not discharged in time. After the solidification of the molten pool, the powder particles are solidified in the weld to form particle slag. As shown in Figure 4h, on the other hand, the particle size is different. Large particles need more heat to melt. The laser is characterized by fast heating speed and fast cooling speed. The laser cannot melt particles with a larger diameter at one time, and the particles with a smaller diameter have a large specific surface area, a large heating area per unit, and high heat absorption. During laser additive process, the small particles will first melt and stick around the large particles to further block the transmission of laser heat. Finally, the large particles and the small particles below the large particles cannot be completely melted to form defects, as shown in Figure 4f.

### 3.3. Surface Fatigue Damage Analysis

Figure 5 shows the crack propagation morphology on the surface of SLM-316L specimens under different stress conditions. When the stress is 250 MPa, the main crack begins on the surface of the sample, and then expands in a Z-shape way under cyclic loading. Holes are found on the surface of the sample. These holes are the weak areas in the process of crack growth, where the tip of the crack tends to propagate. Through the observation of the overall propagation path of cracks in Figure 5a, we found that multiple deflections had occurred during the propagation, and the secondary microcracks that were propagating along the surface defects of the specimen are differentiated at I, II and III in Figure 5a. Among them, the microcracks at I and II no longer propagated forward after extending a certain distance, while the main cracks at III were divided into microcracks that propagated in two different directions. Careful observation shows that when crack 1 did not extend to the microcracks at position III, secondary microcracks had been generated around the holes. These microcracks shared the crack driving force at the tip of crack 1 and reduced the driving force at crack 1, making this insufficient for crack 1 to continue to expand and therefore rendering crack 1 blunt. At this time, crack 1 and the defects in the upper left corner of the specimen had formed a tip plastic zone in a certain area. With the further expansion of the crack, crack 1 may have expanded to the defects in the upper left corner. Further amplification of the crack origin is shown in Figure 5d. Visibly, there are pore defects holes on the surface of the specimen. During the fatigue test, the stress concentration occurs at the hole, resulting in the initiation of microcracks. Obvious slip can be found at the edge of fracture. From Figure 5e,i, it can be seen that when the cyclic stress is 270 MPa and 290 MPa, respectively, the main cracks of SLM-316L samples originated from the surface of the samples and the number of deflections during the process of crack propagation was more than that of 250 MPa. In addition, slip can be found near the crack of all the fracture samples.

Figure 6 shows the fatigue damage of the sample surface after the fatigue test of SLM-316L was interrupted under different stresses. Figure 6a shows the plastic damage surface of the sample when the maximum cyclic stress was 250 Mpa. Visibly, there is a plastic zone at the crack tip, and a certain part of the crack tip has been enlarged and is visible in Figure 6b. From Figure 6b, we can see a lot of dislocation slip bands with a slip distance of 0.5–1 μm, and different types of slip are marked. Type I slip is distributed regularly in a cluster, and this slip originates from the grain boundaries. Type II slip exists in isolation, and this slip is often located close to the grain boundary. Type III slip is the formation of very small slip marker clusters inside the grains and near the grain boundaries. Two sliding mark directions are usually observed. The first one is more significant and occupies most of the surface of the cluster. This is probably related to the slip zone of the main slip system. The second one is less obvious, and is called the secondary slip zone [27].As shown in Figure 6a,b, we can see that a large amount of type I slip exists, and this slip originates from the grain boundary. In Figure 6a, we can see the existence of type II slip. Type II slip exists in isolation and is not distributed in a certain regular cluster, like type I slip. This type of slip is often located near the grain boundary. No obvious type III slip was found. Figure 6c shows the plastic damage surface of the sample when the maximum cyclic stress is 270 MPa, and the crack tip is enlarged to obtain Figure 6d. From Figure 6d, a large number of dislocation slip bands can also be seen. Compared with the stress of 250 MPa, only I-type slip is found, and the distance is similar to the stress of 250 MPa. Figure 6e shows the plastic damage surface of the sample when the maximum cyclic stress is 290 MPa. Figure 6f is obtained by enlarging a certain part of the crack tip. In Figure 6f, a large number of dislocation slip bands can be seen. Compared with 250 MPa stress, the distance is similar to 250 MPa stress. From Figure 6e, we can also find Type I slip, however the amount of Type I slip is reduced. In addition to Type I slip, there is also Type II slip. The slip pattern is affected by the material properties, stress and frequency during fatigue. The slip form is influenced by material properties, fatigue stress and frequency. Different types of slip are only affected by the stress amplitude. Under higher stress amplitude, the number of type I slip decreases and the number of type II slip increases with the decrease in stress amplitude. At very low stress amplitudes, type III slip dominates, while type I slip rarely occurs. When the early type I slip occurs on the surface of the sample, more parallel slip bands are formed inside the grain with the increase in the number of cycles. For type II slips, most are formed at the twin boundary. For type III slip, the amount of slippage in the early stage is very small, and during the cyclic loading process, the slip also changes to type I and type II. The existence of slip is caused by the plastic deformation of the sample during the fatigue test. In the loading process of the sample subjected to cyclic stress, a lot of dislocation movements occur inside the sample, a large number of dislocation movements produce plastic deformation, and the dislocation movement is along a certain direction and that of the slip plane.

## 4. Discuss

### 4.1. Fatigue Crack Initiation Mechanism

We can see from Figure 5 that the crack originates from the sample surface, and defects such as holes are found at the crack source of the sample. Figure 6 shows that a large number of slip bands exist at the crack tip, and the fracture mechanism of the sample can be reflected through the analysis of the pores and slip bands. In this paper, the corresponding model diagram was established to analyze the crack formation of the sample, as shown in Figure 7. Prior to the fatigue test, the surface of the sample was smooth, without obvious unevenness. When the fatigue loading started, a lot of slip bands shown in Figure 6 indicated that some areas on the sample surface would slip during cyclic loading, resulting in the resident slip band, which is a dislocation wall formed by dislocation. With the continuous fatigue loading, a lot of non-uniform regions formed by intrusive extrusion are generated on the surface of the sample. As shown in Figure 7a, with the continuous cyclic loading, slip occurs inside the sample. In the process of actual fatigue loading and reverse loading, material properties change, and this results in the slip that does occur not being completely reversible. Intrusive extrusion steps are generated at some positions on the surface of the SLM-316L sample, which makes the sample surface no longer smooth and forms an uneven area. With further fatigue loading, as shown in Figure 7b, this uneven plastic deformation region is prone to causing stress concentration during cyclic loading, which promotes the initiation of microcracks. During the SLM process, the molten pool exists for a short time and the fluidity of the molten metal decreases. Warpage deformation, tensile residual stress, porosity, unmelted powder, and unfused (LOF) defects are inevitable [28]. The formed 316L sample may have defects such as holes inside or on the surface. According to Griffith’s theory, the stress value in the absence of holes is one third of that in the presence of circular holes. Under a certain stress, with the continuous fatigue loading, microcracks will be easily generated around the holes and other defects due to stress concentration, and the existence of stress concentration will also affect the propagation of cracks. In the actual fatigue loading process, as shown in Figure 5, the defects on the surface and inside of the specimen are mostly irregular shapes, which make the stress concentration coefficient at these defects larger, and microcracks tend to initiate at these locations. As shown in Figure 7c, the microcracks initiate in multiple defects and continue to expand under the action of fatigue load. The expanded microcracks will connect with each other to form the main crack, and ultimately make the specimen fracture in advance.

### 4.2. Characteristics of Fatigue Crack Growth

In general, under certain stress, the main crack generated by fatigue loading extends along the direction perpendicular to the load. However, due to the influence of impurities, holes and other defects, the crack can propagate in different directions. The research of Beachem and Yoder studied crack propagation [29]. After the slip system at a certain position of the specimen is activated to produce microcracks, the tip of the microcrack will continue to undergo passivation, making its propagation path appear Z-shaped. In this paper, the crack propagation characteristics of the sample with defects are analyzed through the model. As shown in Figure 8a, when the stress is low, there is little possibility of microcracks occurring at the defect near the initiation of the main crack. When the stress is high, the defects near the main crack initiation zone will also produce microcracks, and the microcracks will expand a certain distance under the action of tensile stress, and some cracks may bridge with the main cracks. During the forward propagation of the main crack, a certain angle deflection always occurs to the area where the defect exists, as shown in Figure 8c,d. Many bridging areas are formed at the tip of the main crack and at the defect near the tip, which is more conducive to the accelerated propagation of fatigue crack.

## 5. Conclusions

(1) The surface morphologies of SLM-316L stainless steel samples with different forming directions are different. In the direction perpendicular to the laser additive manufacturing, the melting tracks of the sample are well bonded; In the sample parallel to the additive manufacturing direction, a lot of fish scale melting tracks can be found, which are superimposed together. The surface of these tracks adheres to a lot of unmelted metal powder, and there are small holes in the overlap between the tracks.

(2) Fatigue tests of SLM-316L and rolled 316L were carried out 1 × 10^6^ times under different stresses. Due to the existence of defects, the fatigue performance of SLM-316L specimen is inferior to that of rolled 316L specimen. The fatigue fracture surface of SLM-316L sample was smooth, and no obvious plastic deformation traces were found. The fracture mode was brittle fracture. However, there are radiation patterns at the fatigue source of rolled 316L, and clear dimples can be observed on the fracture surface. The fracture mode is ductile fracture.

(3) In the process of cyclic loading, some areas on the surface of SLM-316L sample slipped to produce resident slip bands, and a lot of non-uniform areas formed by intrusive extrusion were generated on the surface of the sample. The existence of hole defects promoted the initiation of microcracks, and notably, they also accelerated the propagation of cracks.

(4) After the fatigue microcracks of SLM-316L sample were generated, the crack tip expanded along the front defect due to the influence of external load, which was generally in a Z-shaped way. The inevitable defects during the SLM process affected the propagation of fatigue within the main crack in the form of bending, bifurcation and bridge, which always deviates from the maximum load direction in the process of main crack propagation.

## Figures and Tables

**Figure 1 materials-14-07544-f001:**
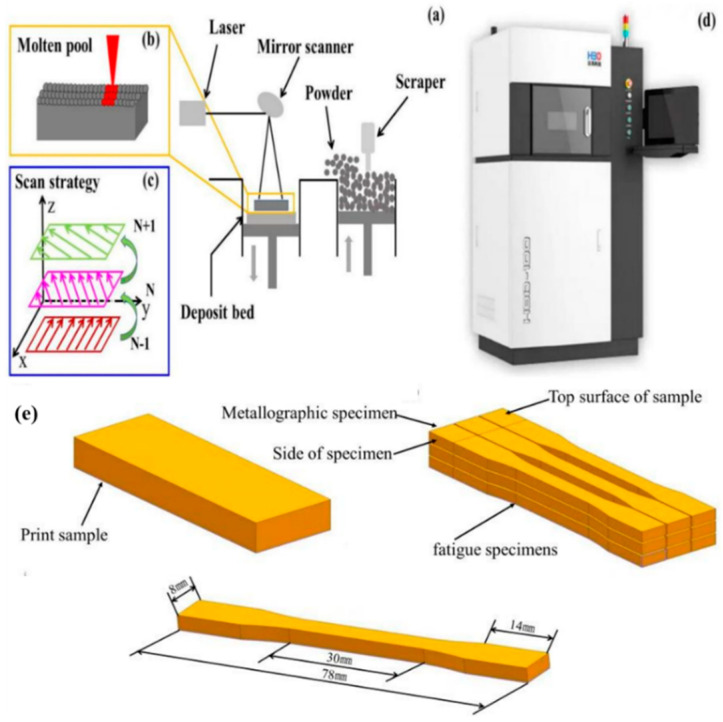
SLM forming equipment HBD-100D and cutting diagram of metallographic and fatigue samples. (**a**) Printing process, (**b**) molten pool, (**c**) Printing strategy, (**d**) Print instrument, (**e**) cutting diagram of metallographic and fatigue sam-ples.

**Figure 2 materials-14-07544-f002:**
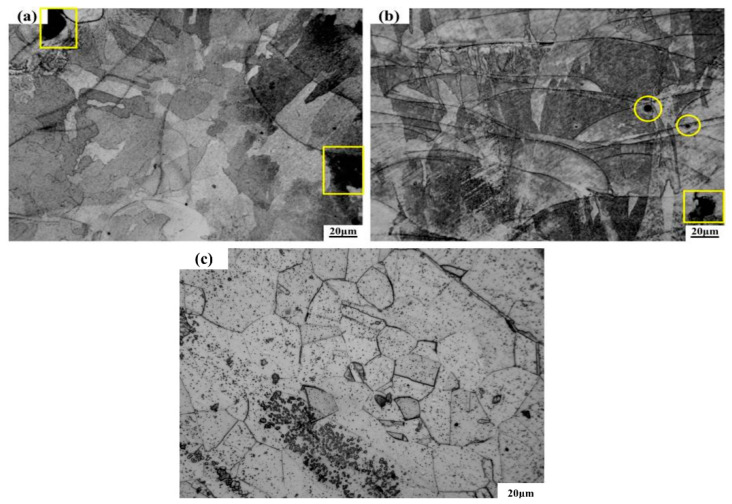
Microstructure of SLM-316L stainless steel samples in different building directions and rolling 316L: (**a**) microstructure of SLM-316L stainless steel sample perpendicular to the building direction, (**b**) microstructure of SLM-316L stainless steel sample parallel to the building direction, (**c**) microstructure of rolling 316L.

**Figure 3 materials-14-07544-f003:**
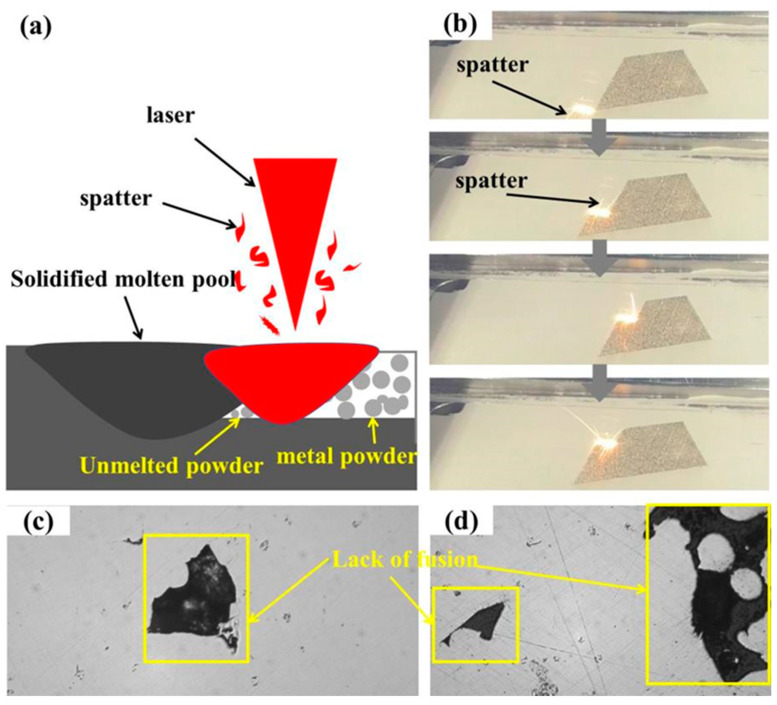
Mechanism of spatter and incomplete fusion in SLM forming process: (**a**) laser scanning single channel metal powder model, (**b**) forming process, (**c**) lack of fusion, (**d**) lack of fusion with unmelted metal powder.

**Figure 4 materials-14-07544-f004:**
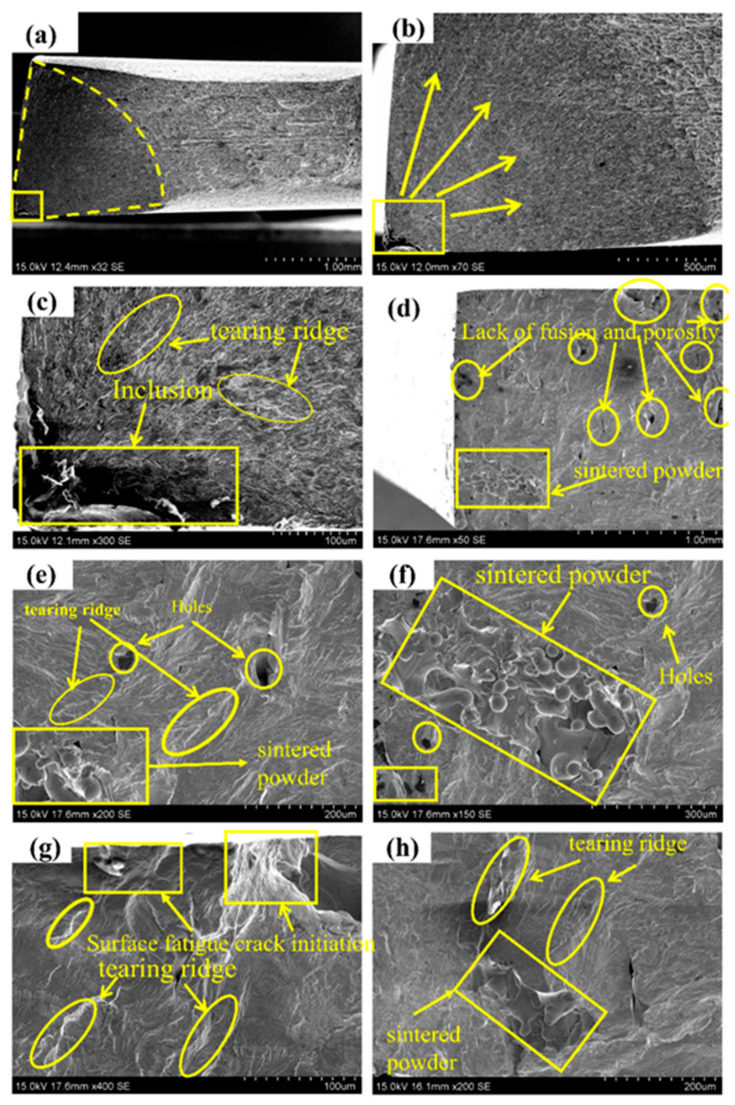
Fracture morphology of rolzSLM-316L fatigue initiation: (**a**–**c**) rolling 316L, (**d**–**h**) SLM-316L.

**Figure 5 materials-14-07544-f005:**
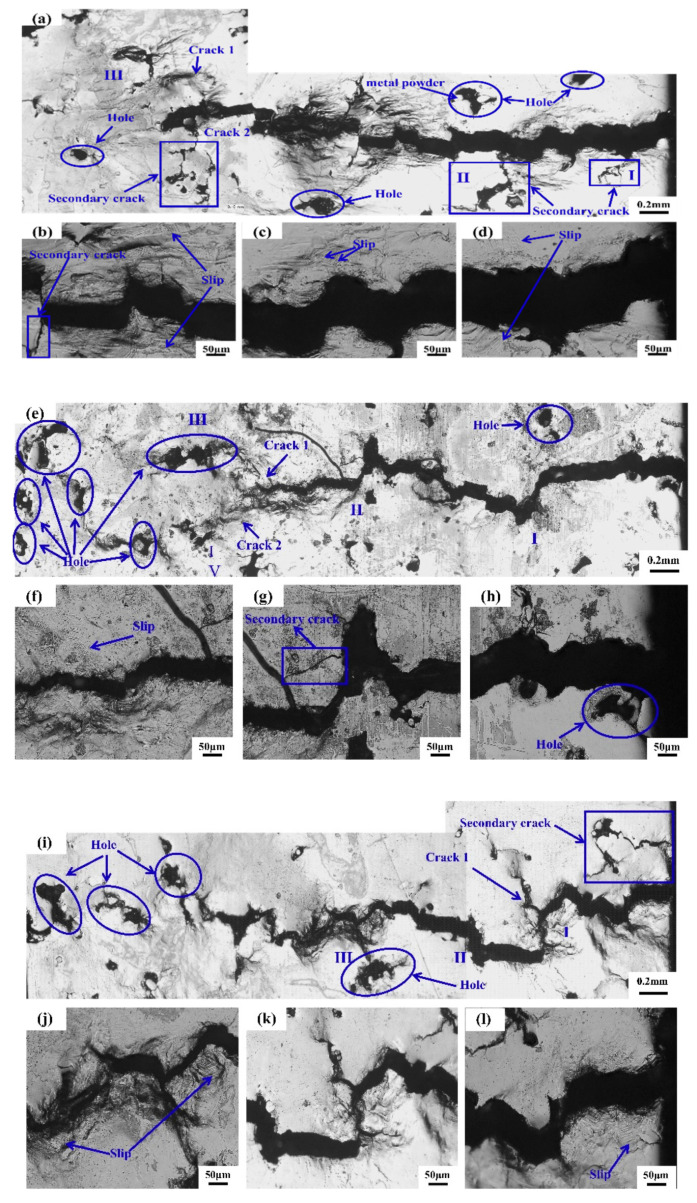
Surface crack growth of SLM-316L under different stresses: (**a**–**d**) 250 MPa, (**e**–**h**) 270 MPa, (**i**–**l**) 290 MPa.

**Figure 6 materials-14-07544-f006:**
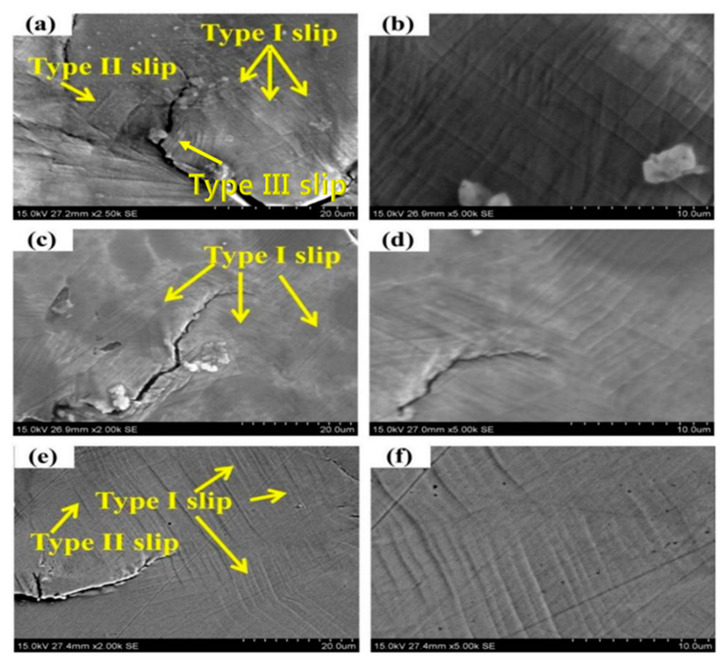
Surface fatigue damage morphology of SLM-316L under different stresses: (**a**,**b**) 250 MPa, (**c**,**d**) 270 MPa, (**e**,**f**) 290 MPa.

**Figure 7 materials-14-07544-f007:**
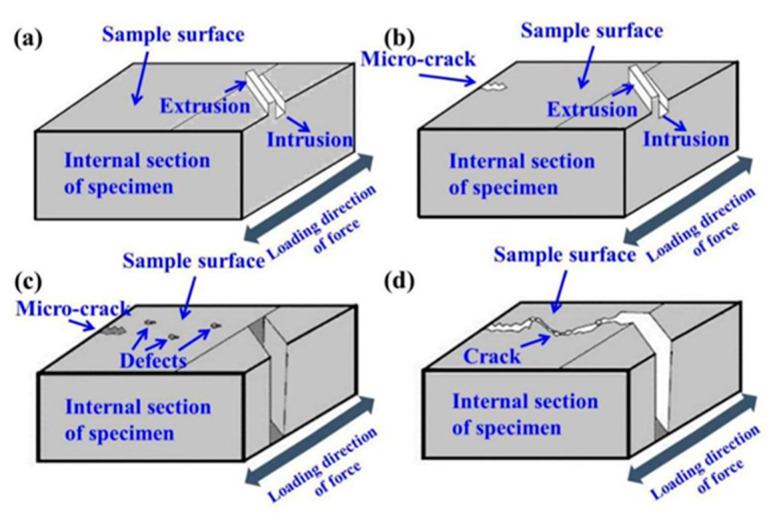
Schematic diagram of fatigue crack formation mechanism; (**a**) Schematic diagram of slip generation, (**b**) Schematic diagram of microcrack generation, (**c**) Schematic diagram of crack growth, (**d**) Schematic diagram of crack propagation path.

**Figure 8 materials-14-07544-f008:**
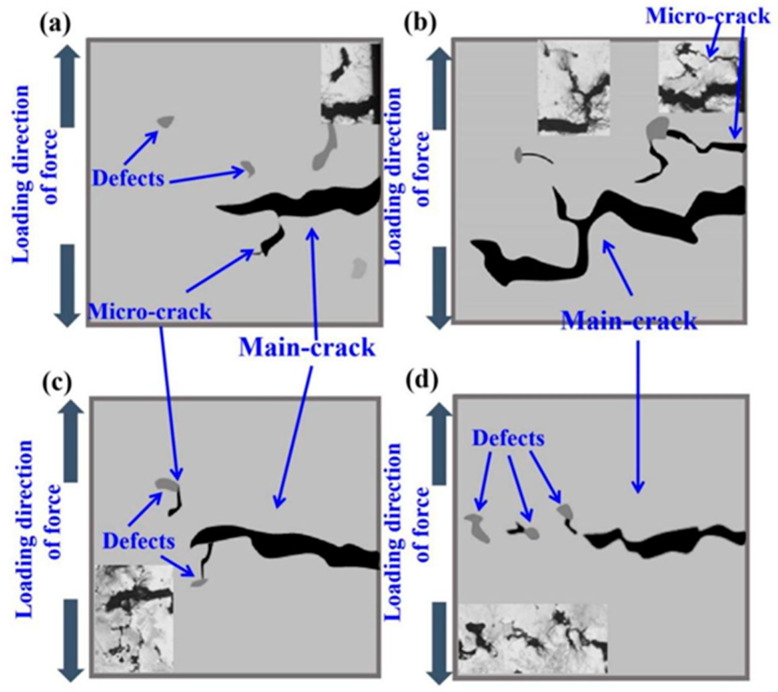
Sketch of fatigue crack propagation characteristics in the presence of defects; (**a**,**b**) Schematic diagram of main defect expansion, (**c**,**d**) Schematic diagram of the effect of defects on crack propagation.

**Table 1 materials-14-07544-t001:** Composition of powder and rolled 316L.

Composion	Cr	Mn	Ni	Mo	Cu	C	P	S	Fe
316L powder	16.97	0.8	12.2	2.89	0.01	0.003	0.002	0.006	Bal.
Rolled 316L	16–18	≤2.0	10–14	2.0–3.0	----	≤0.03	≤0.045	≤0.03	Bal.

**Table 2 materials-14-07544-t002:** SLM process parameters.

Materials	Scanning Distance (mm)	Laser Power (W)	Scanning Speed (mm/s)	Layer Thickness (μm)	Spot Distance (μm)
316L	0.07	160	1000	30	70

**Table 3 materials-14-07544-t003:** Fatigue test data of rolled 316L and SLM-316L.

Type	Sample No.	Maximum Stress (Smax/Mpa)	Stress Amplitude (Sa/Mpa)	Life Cycle
Rolled	B-1	460	207	241,870
Rolled	B-2	440	198	262,350
Rolled	B-3	420	189	354,397
Rolled	B-4	400	180	461,752
Rolled	B-5	380	171	600,691
Rolled	B-6	370	166.5	1,000,000
Rolled	B-7	360	162	1,000,000
SLM	S-1	300	135	421,906
SLM	S-2	280	126	399,303
SLM	S-3	260	117	465,199
SLM	S-4	240	108	454,712
SLM	S-5	220	99	530,287
SLM	S-6	210	94.5	580,493
SLM	S-7	200	90	1,000,000

## Data Availability

The data used to support the findings of this study are included within the article.

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
