# Peer review of "Microstructure and Fatigue Damage of 316L Stainless Steel Manufactured by Selective Laser Melting (SLM)"

_materials, 2021, doi:10.3390/ma14247544_

Round 1

Reviewer 1 Report

The paper proposed by the authors deals with the investigation of the mechanism of fatigue crack initiation and propagation of a AISI 316L stainless steel produced by additive manufacturing. First, constant amplitude fatigue tests were carried out on AISI 316L stainless steel specimens produced by additive manufacturing and specimens made of the same material produced by conventional manufacturing process. Secondly, the crack paths and their interaction between process-inherent defects were investigated. 

The paper's topic is not fully studied in the literature so this work is consistent with the motivation expressed by the authors in the introduction. 

However, the paper requires extensive editing of English language and style. The reviewer recommends a deep review of the English language by a native English speaker.

Some further comments related to the contents of the paper are reported in the following: 

  • page 2 line 5: the authors wrote "However, there are few reports about the fatigue failure mechanism of SLM-316L, and fatigue failure is one of the most common failure forms of structural components under alternating loads" by citing only ref. [17]. According to the reviewer knowledge, there are several works that deals with fatigue of AISI 316L produced by AM. Please, expand the literature review in the introduction.
  • page 2 line 5: a footnote is cited in the manuscript but it is not clear what is referred to. 
  • page 3 line 8: ref. [18] is not pertinent with the fatigue of AISI 316L and it is in contrast to what is written in the manuscript. Please check the references of the manuscript.
  • page 3 line 2. "During the printing, it rotated by 67° layer by layer" unclear sentence.
  • table 2: Arabic fonts should be used in international journals.
  • page 4 line 2. The direction of additive manufacturing is not defined. Normally, it is called "build direction"
  • Page 5: What is "group test method"
  • Page 6 line 2: "times" or "cycles"
  • Page 9 "the first type....sample surface". Figs. 6d and 6f are not cited in the manuscript. The explanation of the slip type is not clear. Please modify the explanation. 
  • Page 9 "The slip pattern ... fatigue stress, load frequency": is this a conclusion of this work or such behaviour is already reported in the literature about both AM and conventionally manufactured stainless steels? 
  • page 11 please check this sentence "In addition, due to the inevitable selective laser melting process, the formed 316L sample may have defects such as holes inside or on the surface."

Author Response

Thank you for your valuable comments on my article, the following is the answer to your suggestions.

  • 1、page 2 line 5: the authors wrote "However, there are few reports about the fatigue failure mechanism of SLM-316L, and fatigue failure is one of the most common failure forms of structural components under alternating loads" by citing only ref. [17]. According to the reviewer knowledge, there are several works that deals with fatigue of AISI 316L produced by AM. Please, expand the literature review in the introduction.

( After modification)

However, there are few reports about the fatigue failure mechanism of SLM-316L, and fatigue failure is one of the most common failure forms of structural components under alternating loads[17-20]. The research on fatigue failure of SLM-316L is of great significance to the development of SLM technology.

Response 1:

In addition to the literature [17], this article cite three more literatures [18][19][20].

[17] Uhlmann E, Fleck C, Gerlitzky G, et al. Dynamical Fatigue Behavior of Additive Manufactured Products For a Fundamental Life cycle Approach[J]. Procedia Cirp, 2017, 61:588-593.

[18] Zhang C ,  Cao D ,  You W , et al. Fatigue failure of welded details in steel bridge pylons[J]. Engineering Failure Analysis, 2021, 127(3):105530.

[19] Sajith S ,  Shukla S S ,  Murthy K , et al. Mixed mode fatigue crack growth studies in AISI 316 stainless steel - ScienceDirect[J]. European Journal of Mechanics - A/Solids, 80

[20] He L ,  Akebono H ,  Sugeta A . Effect of high-amplitude loading on accumulated fatigue damage under variable-amplitude loading in 316 stainless steel[J]. International Journal of Fatigue, 2018, 116(NOV.):388-395.

  • 2、page 2 line 5: a footnote is cited in the manuscript but it is not clear what is referred to. 

Response 2:

The footnote in the fifth line of the second page is an error. The footnote has been deleted.

3、page 3 line 8: ref. [18] is not pertinent with the fatigue of AISI 316L and it is in contrast to what is written in the manuscript. Please check the references of the manuscript.

Response 3:

The reference [18] in the article has been revised, and an article titled "On the fatigue crack growth behavior in 316L stainless steel manufactured by selective laser melting" is re-quoted in the article.

Riemer et al The fatigue behavior of 316L manufactured by selective laser melting is used to check the crack initiation and crack growth behavior[21]. The results obtained clearly show that 316L is a promising candidate for cyclically loaded parts manufactured by SLM. Primarily attributed to the high ductility directly following SLM processing, the 316L stainless steel shows fatigue properties similar to conventionally processed material in its as-built. ( After modification)

  • page 3 line 2. "During the printing, it rotated by 67° layer by layer" unclear sentence.

There is a schematic diagram of laser additive in the article, but it is not quoted. It has been corrected and quoted.

Response 4:

Fig. 1 SLM forming equipment HBD-100D and cutting diagram of metallographic and fatigue samples

In the printing process, rotate 67° layer by layer, using the scanning strategy shown in Fig.1c.(After modification)

  • 5、table 2: Arabic fonts should be used in international journals.

Response 5:

Material

Scan pitch(mm)

Laser power(W)

Scan speed (mm/s)

Layer thickness(μm)

Spot diameter(μm)

316L

0.07

160

1000

30

70

6、page 4 line 2. The direction of additive manufacturing is not defined. Normally, it is called "build direction"

Response 6:

Fig.2a and b respectively show the metallographic microstructure of samples in the vertical and parallel directions of additive manufacturing.(original)

The sample is based on the XOY surface as the bottom surface and the Z axis as the construction direction, as shown in Fig.1c.Fig.2a and b respectively show the metallographic microstructure of samples in the vertical and parallel directions of construction direction.(After modification)

  • Page 5: What is "group test method"

Response 7:

The group test method is a writing error, and the correct method is the rise and fall method.

In this paper, the group test method was used to carry out 106 cycle fatigue tests on rolled 316L and SLM-316L,(original)

In this paper, a laboratory hydraulic servo low-frequency fatigue testing machine is used to apply a load to simulate the actual service process of the sample. Follow the national standard GB/T 3075-2008 "Metal Material Fatigue Test Axial Force Control Method", using the lifting method to perform the fatigue test with 106 cycles for rolling 316L and laser selective additive manufacturing 316L,(After modification)

8、Page 6 line 2: "times" or "cycles"

Response 8:

Modifications to times

  • Page 9 "the first type....sample surface". Figs. 6d and 6f are not cited in the manuscript. The explanation of the slip type is not clear. Please modify the explanation.

Response 9:

Fig. 6d and Fig.6f are quoted in the text after the amendment, and the description of the slip part is modified.

Fig.6 shows the fatigue damage of the sample surface after the fatigue test of SLM-316L is interrupted under different stresses. Fig.6a shows the plastic damage surface of the sample when the maximum cyclic stress is 250Mpa. It can be seen that there is a plastic zone at the crack tip, and a certain part of the crack tip is enlarged to get Fig.6b. From Fig. 6b, we can see a large number of dislocation slip bands with a slip distance of 0.5-1μm, and different types of slip are marked. As shown in Figures 6a and 6b, it can be seen that a large amount of type I slip exists, and this slip originates from the grain boundary. In Fig.6a, you can see the existence of type II slip. Type II slip exists in isolation. It is not distributed in a certain regular cluster like type I slip. This type of slip is often located near the grain boundary. No obvious type III slip was found[24]. Fig.6c shows the plastic damage surface of the sample when the maximum cyclic stress is 270 MPa, and the crack tip is enlarged to obtain Fig.6d. From Fig.6d, a large number of dislocation slip bands can also be seen. Compared with the stress of 250MPa, only I-type slip is found, and the distance is similar to the stress of 250MPa. Fig.6e shows the plastic damage surface of the sample when the maximum cyclic stress is 290 MPa. Fig.6f is obtained by enlarging a certain part of the crack tip. In Fig.6f, a large number of dislocation slip bands can be seen. Compared with 250MPa stress, the distance is similar to 250MPa stress. From Fig.6e, we can also find Type I slip, but the amount of Type I slip is reduced. In addition to Type I slip, there is also Type II slip. (After modification)

10、Page 9 "The slip pattern ... fatigue stress, load frequency": is this a conclusion of this work or such behaviour is already reported in the literature about both AM and conventionally manufactured stainless steels?

Response 10:

Slip mode refers to the slip mode of other materials, and names the slip mode of stainless steel according to the characteristics of the slip form of other laser additive materials.

Fatigue stress is to judge an approximate value of fatigue stress based on the tensile strength of the material, and then use the lifting method to test the fatigue stress range.

Load frequency: This fatigue data is tested with a low-cycle fatigue testing machine. The vibration frequency of the low-cycle fatigue testing machine is adjustable. The vibration frequency of this test is 15 times per second.

Slip mode, fatigue stress, and load frequency are the conclusions of this work.

11、page 11 please check this sentence "In addition, due to the inevitable selective laser melting process, the formed 316L sample may have defects such as holes inside or on the surface."

Response 11:

     In the selective laser melting process, the molten pool exists for a short time and the fluidity of the molten metal decreases. Warpage deformation, tensile residual stress, porosity, unmelted powder, and unfused (LOF) defects are inevitable [29]. the formed 316L sample may have defects such as holes inside or on the surface.( After modification)

12、Extensive editing of English language and style required

Response 12:

The English and grammar in the article have been extensively revised。

Thank you for your valuable suggestions on my article

Reviewer 2 Report

Microstructure and Fatigue Damage of 316L Stainless Steel Manufactured by Selective Laser Melting (SLM) 

Comments:

It is not clear whether the specimens for the fatigue test were printed under the standard or rectangular samples were printed that were subsequently machined.

It would be important for rolling samples of 316L to display the results of the Surface crack growth under different stresses.  Since although it is clear that the rolled samples reach a higher level of stress, if we visualize the life cycles we can see that they are in a very similar average. 

Author Response

1、It is not clear whether the specimens for the fatigue test were printed under the standard or rectangular samples were printed that were subsequently machined.

Response 1:

In this experiment, the fatigue specimen was processed, and a rectangular specimen was manufactured by laser selective area additive and then processed into a fatigue pattern. As shown in Figure e below

2、It would be important for rolling samples of 316L to display the results of the Surface crack growth under different stresses.  Since although it is clear that the rolled samples reach a higher level of stress, if we visualize the life cycles we can see that they are in a very similar average.

Response 2:

When rolling 316L for fatigue test, the difference of each stress level is only 20MPa. The difference in stress levels is small, so the life of the specimens is not much different.

Reviewer 3 Report

Reviewer Recommendation and Comments for manuscript materials-1421088 with the title: “Microstructure and Fatigue Damage of 316L Stainless Steel Manufactured by Selective Laser Melting (SLM)”, authors: Wang Zhentao, Yang Shanglei , Huang Yubao, Fancong, Peng zeng, Gao Zihao.

The authors present 316L stainless steel manufacturing by selective laser melting and its study regarding the fatigue failure and crack initiation and propagation. The equipment used in this study is represented by a HBD-100SLM apparatus; optical microscope and a scanning electron microscope (SEM).

The article may be published after MAJOR REVISION.

The main comments that I find useful for improving the quality of the article are presented below:

#Are the names of all the authors spelled correctly?

#This work is supported by experimental results but the authors do not compare their data with literature.

#Table 2 must be translated to English.

#The compositions of substrate and rolled 316L steel must be introduced into Table 1.

#The experimental data presented in Table 3 need to be explained in more detail.

#Figure 4 needs to be replaced. Other graphics should be used to allow easy visualization of the writing, especially the blue one.

#The correspondence between samples B1 ... B7 and figures 4a ... 4c must be made.

#The correspondence between samples S1 ... S7 and figures 4d ... 4h must be made.

#It is very difficult to notice the writing in Figure 5. Other graphics should be used to allow easy visualization of the writing.

#The authors show in Fig.5 the increase of the surface cracks of SLM-316L under different stresses: (a-d) 250MPa, (e-h) 270MPa, (i-l) 290MPa. These samples are different from samples S1 ... S7. Are the analyzes performed on other samples?

#Comments seem to be intentionally directed in a certain direction. Such a study does not present a statistic on method / efficiency. The authors do not present any evidence of effective coverage. They analyze 7 samples (S1…S7), but do not show any effective test. Are all samples defective? Do all analyzed materials have cracks or pores? The reader can be confused! According to the authors, shouldn't this method be used?

#The Materials journal requires a specific Template that must be used!

#The Materials journal requires a specific format of citing references. In the text, reference numbers should be placed in square brackets [ ] and placed before the punctuation.

#The Materials journal require a specific format of references, authors must pay more attention in their writing. No reference is written according to the format required by the journal.

#There are some grammar and typing mistakes.

#The authors must revise the entire manuscript.

Author Response

Thank you for your valuable comments on my article, the following is the answer to your suggestions.

1、#Are the names of all the authors spelled correctly?

Response 1:

Wang Zhentaoa, Yang Shangleia,b , Huang Yubaoa, Fan conga,Peng zenga, Gao Zihaoa (original)

Zhentao Wanga ,Shanglei Yang a,b , Yubao Huang a, cong Fana, zeng Peng a, Zihao Gaoa (After modification)

2、#This work is supported by experimental results but the authors do not compare their data with literature.

Response 2:

Cited in the article [21] "On the fatigue crack growth behavior in 316L stainless steel manufactured by selective laser melting" is an article on the fatigue and fatigue crack growth of 316L manufactured by selective laser melting, which is in contrast with this article.

3、#Table 2 must be translated to English.

Response 3:

Material

Scan pitch(mm)

Laser power(W)

Scan speed (mm/s)

Layer thickness(μm)

Spot diameter(μm)

316L

0.07

160

1000

30

70

4、#The compositions of substrate and rolled 316L steel must be introduced into Table 1.

     The composition of rolled 316L steel has been quoted in Table 1.

Response 4:

Composion

Cr

Mn

Ni

Mo

Cu

C

P

S

Fe

316L powder

16.97

0.8

12.2

2.89

0.01

0.003

0.002

0.006

Bal.

Rolled 316L

16-18

≤2.0

10-14

2.0-3.0

----

≤0.03

≤0.045

≤0.03

Bal.

5、#The experimental data presented in Table 3 need to be explained in more detail

Response 5:

In this paper, a laboratory hydraulic servo low-frequency fatigue testing machine is used to apply a load to simulate the actual service process of the sample. Follow the national standard GB/T 3075-2008 "Metal Material Fatigue Test Axial Force Control Method", using the lifting method to perform the fatigue test with 106 cycles for rolling 316L and laser selective additive manufacturing 316L,and fatigue data results Are shown in Table 3. According to the data in Table 3, the fatigue performance of laser-selected area additive manufacturing 316L is lower than that of rolled 316L.

6、#Figure 4 needs to be replaced. Other graphics should be used to allow easy visualization of the writing, especially the blue one.

Fig.4 Fracture morphology of rolling 316L and SLM-316L fatigue intiation: (a-c) rolling 316L, (d-h) SLM-316L(After modification)

7、#The correspondence between samples B1 ... B7 and figures 4a ... 4c must be made

Response 7:

Fig.4(a-c) shows the fracture morphology of the rolled 316L B-5 group sample. The stress level of the sample is 380MPa and the fatigue life is 600691 . (After modification)

8、#The correspondence between samples S1 ... S7 and figures 4d ... 4h must be made.

Response 8:

Fig.4(d-f) shows the fracture morphology of the SLM 316L S-6 group samples, the stress level is 210MPa, and the fatigue life is 580493.

Response 9:

9、#It is very difficult to notice the writing in Figure 5. Other graphics should be used to allow easy visualization of the writing.

10、#The authors show in Fig.5 the increase of the surface cracks of SLM-316L under different stresses: (a-d) 250MPa, (e-h) 270MPa, (i-l) 290MPa. These samples are different from samples S1 ... S7. Are the analyzes performed on other samples?

Response 10:

When observing surface cracks and surface crack propagation, we need unbroken samples. S1-S7 are not removed from the fatigue testing machine until the specimen is broken. Such specimens cannot observe the growth of surface crack tips, nor can they observe surface cracks. After we observe the fractures of the S1-S7 samples, there is no big difference in the number of deflection of the main crack below 240MPa. The smaller the stress, the denser slip band on the surface is not conducive to observation. Therefore, three stress levels of 250 MPa, 270 MPa, and 290 MPa are selected for the test.

11、#Comments seem to be intentionally directed in a certain direction. Such a study does not present a statistic on method / efficiency. The authors do not present any evidence of effective coverage. They analyze 7 samples (S1…S7), but do not show any effective test. Are all samples defective? Do all analyzed materials have cracks or pores? The reader can be confused! According to the authors, shouldn't this method be used?

Response 11:

When analyzing the laser selective area additive manufacturing 316L samples, each sample has some defects, and you can also observe the defects by observing the metallographic picture of the sample. When observing the fracture of the material, there are cracks and gaps in the fracture of each sample. The main purpose of this article is two points. The first point is to perform fatigue tests on the 316L sample and the rolled 316L sample manufactured by laser selective additive manufacturing with the process parameters in Table 1, and compare their fatigue properties. The second point: study the cause and process of fatigue damage of 316L specimens manufactured by laser selective area additive manufacturing in table 1 process parameters.

Table.1 SLM process parameters

Material

Scan pitch(mm)

Laser power(W)

Scan speed (mm/s)

Layer thickness(μm)

Spot diameter(μm)

316L

0.07

160

1000

30

70

12、#The Materials journal requires a specific Template that must be used!

Response 12:

The article has been revised according to the journal template.

13、#The Materials journal requires a specific format of citing references. In the text, reference numbers should be placed in square brackets [ ] and placed before the punctuation.

Response 13:

The literature has been revised in accordance with the journal template.

14、#The Materials journal require a specific format of references, authors must pay more attention in their writing. No reference is written according to the format required by the journal

Response 14:

The literature has been revised in accordance with the journal template.

15、#There are some grammar and typing mistakes.

Response 15:

The grammar of the article has been modified

16 #The authors must revise the entire manuscript

Response 16:

The English and grammar of the article were carefully checked.

Thank you for your valuable suggestions on my article

Round 2

Reviewer 1 Report

Dear authors, 

after corrections, the paper can be published in the present form. 

Best regards

Author Response

Hello, reviewer, the article has been corrected according to your suggestions, thank you for your suggestions on the article.

Reviewer 3 Report

Reviewer Recommendation and Comments for manuscript materials-1421088 with the title: “Microstructure and Fatigue Damage of 316L Stainless Steel Manufactured by Selective Laser Melting (SLM)”, authors: Zhentao Wang, Shanglei Yang, Yubao Huang, cong Fan, zeng Peng, Zihao Gao.

The authors partially correct the manuscript. In some places, the authors make irrelevant changes with synonymous words (large / lot), (Fig./Figure). There are some important questions that have not been answered.

The article may be published after MINOR REVISION.

In part, the article is improved, but I believe that the following observations may bring a better version of the manuscript. The main comments that I find useful for improving the quality of the article are presented below:

#This work is supported by experimental results but the authors do not compare their data with literature (in Results section).

#Table 1 name is improper. Should be presented both compositions, powder and rolled. (The composition of powder and rolled 316L). Not only powder.

#Table 1. Please change Composion to Composition.

#Please check the typo of Table 2.

#The correspondence between samples B1 ... B7 and figures 4a ... 4c must be made.

#The correspondence between samples S1 ... S7 and figures 4d ... 4h must be made.

#The authors show in Fig.5 the surface cracks of SLM-316L under different stresses: (a-d) 250MPa, (e-h) 270MPa, (i-l) 290MPa. These samples are different from samples S1 ... S7. Are the analyzes performed on other samples? Why these samples are not presented in Table 3?

#Comments seem to be intentionally directed in a certain direction. Such a study does not present a statistic on method / efficiency. The authors do not present any evidence of effective coverage. They analyze 7 samples (S1…S7), but do not show any effective test. Are all samples defective? Do all analyzed materials have cracks or pores? The reader can be confused! According to the authors, shouldn't this method be used?

#The Materials journal require a specific format of references, authors must pay more attention in their writing. No reference is written according to the format required by the journal.

#There are some grammar and typing mistakes.

#Please correct the typos.

#The authors must revise the entire manuscript.

Author Response

#This work is supported by experimental results but the authors do not compare their data with literature (in Results section).

Response 1:

In the results part of this article, a total of 2 documents are cited [26], [27]. As a near-Gaussian heat source, laser has high energy density in the center and small edges. During the laser scanning process, the molten pool is unstable, which easily leads to incomplete fusion of the weld bead and the overlap of the weld bead. This conclusion comes from Literature 26, and the purpose of this conclusion is to explain the causes of laser defects. Not for comparison with article data.

This article changes the 27 parts of the document.

This article first quotes the slip naming and slip characteristics of Reference 27.(Type I slip is distributed regularly in a cluster, and this slip originates from grain boundaries. Type II slip exists in isolation, and this slip is often located close to the grain boundary. Type III slip forms very small clusters of slip marks inside the grains and near the grain boundaries, and two sliding mark directions are usually observed. The first one is more pronounced and occupies most of the surface of the cluster. It is probably related to the slip zone of the main slip system. The second one is less obvious and is called the secondary slip zone. )After that, this article named the slip according to the slip characteristics of the experimental sample and the naming method of literature 27, and compared with literature 27 at the same time.

#Table 1 name is improper. Should be presented both compositions, powder and rolled. (The composition of powder and rolled 316L). Not only powder.

Response 2

The title of Table 1 has been changed.

Table1 Composition of powder and rolled 316L(After modification)

#Table 1. Please change Composion to Composition.

Response 3

already edited

#Please check the typo of Table 2.

Response 4

The errors in Table 2 have been corrected.

Table.2 SLM process parameters

Materials

Scanning distancemm

Laser powerW

Scanning speed mm/s

Layer thicknessμm

Spot distanceμm

316L

0.07

160

1000

30

70

(After modification)

#The correspondence between samples B1 ... B7 and figures 4a ... 4c must be made.

Response 5

sorry. I did not fully understand your comment, but I have answered your comment. If you are not satisfied with my answer, please contact me again. Thanks for your comment.

(After modification)Figures 4a-c are the fracture morphologies of rolled 316 B5, and Figure 4a shows the macroscopic morphology of sample B5. It can be seen that there are fewer internal defects, obvious plastic deformation in the fracture, and obvious inclusion defects in the lower left corner of the fracture. The inclusion defects are located on the surface of the sample and spread radially around. Figure 4b shows an enlarged view of the fatigue source area of sample B5, which is also a further enlargement of Figure 4a. It can be seen that fatigue cracks originate from inclusion defects on the surface of the sample. After the crack is formed, it starts from the defect and spreads radially around. Defects under cyclic loading will cause the stress concentration to be higher than the surroundings, resulting in microcracks in the sample without a large external load. Fig. 4c is an enlarged view of the defect of the fracture surface of sample B5, which is also a further enlargement of Fig. 4b. It can be seen the existence of the tear ridge (inside the yellow ellipse), which is left when the cracks in the expansion process meet when they expand in all directions.

#The correspondence between samples S1 ... S7 and figures 4d ... 4h must be made.

Response 6

sorry. I did not fully understand your comment, but I have answered your comment. If you are not satisfied with my answer, please contact me again. Thanks for your comment.

(After modification)Observe Figure 4d-f, Figure 4d-f is the fracture morphology of SLM-316L sample S6. Figure 4d shows the macro morphology of sample S6. From the figure, it can be seen that there are defects such as sintered powder (in the yellow frame) and unfused pores (in the yellow circle) inside the sample. 4e-h is an enlarged view of the fracture morphology of sample S6, which is also a further enlargement of Figure 4d. As shown in Figure 4e, you can see the tearing ridges produced by the cracks on the left that meet the defects caused by unfusion. The cracks will expand further after encountering the defects during the propagation process. And as shown in Figure 4f, around the sintered powder, the cracks generated will expand to the surroundings. As shown in Fig. 4g, cracks are generated on the surface of the sample, and then expand forward, and tear ridges will also be generated when intersecting with other cracks.

#The authors show in Fig.5 the surface cracks of SLM-316L under different stresses: (a-d) 250MPa, (e-h) 270MPa, (i-l) 290MPa. These samples are different from samples S1 ... S7. Are the analyzes performed on other samples? Why these samples are not presented in Table 3?

Response 7

In this paper, when selecting the stress level of the surface crack propagation specimens, the fractures of the S1-S6 specimens were observed. Below 260Mpa (S3-S6), the slip zone near the fatigue fracture is too dense and difficult to observe, and the macroscopic morphology of the crack is not obvious. Above 260Mpa (S1-S2), the stress level is larger and the slip zone is far apart and easy to observe, and the cracks are also different in the macroscopic morphology. Therefore, this paper decided to choose one sample at 260Mpa and two samples above 260Mpa to observe the effect of stress on the growth of surface cracks. In order to increase the abundance of data, this paper did not choose the three stress levels of 260Mpa, 280Mpa, and 300Mpa for surface crack observation testing, but chose the three stress levels of 250Mpa, 270Mpa, and 290Mpa for surface crack testing. Observe that there is no obvious change between the three groups of stress levels (250Mpa, 260Mpa), (270Mpa, 280Mpa), (290Mpa, 300Mpa), the crack expansion and slip types in the early stage. Therefore, this article only elaborates on the surface crack growth of the three stress levels of 250Mpa, 270Mpa, and 290Mpa. The samples of (S1-S7) are not explained.

When the samples with three stress levels of 250Mpa, 270Mpa, and 290Mpa are subjected to fatigue test, their distance limit is designed to be 1mm. (That is, after the crack propagates for a certain distance, the fatigue test is terminated when the sample has not broken.) After the test is terminated, observe the macro morphology of the crack, the type of slip around the crack and the front end of the crack. For samples with three stress levels of 250Mpa, 270Mpa, and 290Mpa, the samples did not break, and there was no fatigue life data.

#Comments seem to be intentionally directed in a certain direction. Such a study does not present a statistic on method / efficiency. The authors do not present any evidence of effective coverage. They analyze 7 samples (S1…S7), but do not show any effective test. Are all samples defective? Do all analyzed materials have cracks or pores? The reader can be confused! According to the authors, shouldn't this method be used?

Response 8

The SLM-316L sample used in the test in this paper has many defects. Because the defects have a greater impact on the fatigue life, 7 samples in the sample cannot be used as valid data to compare with the data of rolled 316L due to the effect of defects. Therefore, statistics on efficiency or methods cannot be provided. The main purpose of this article is to analyze the structure and fatigue damage of laser selective area additive 316L. This article does not conduct a detailed study on the laser selective area additive parameters, but uses the existing parameters for laser selective area additive manufacturing of 316L, which cannot guarantee that the quality of the samples is the highest. This article has introduced the unavoidable defects in the laser selective area additive process, and cited literature to prove it. In the fatigue test, the defects are most likely to cause cracks. Therefore, observe the macroscopic fracture morphology of the samples (S1-S7), and each of them has more or less defects. All samples in this article are manufactured with the same parameters, which can only show that the fatigue performance of SLM-316L under this parameter is lower than that of rolled 316L. It cannot be proved that the fatigue performance of 3L6L printed by SLM is lower than that of rolled 316L. In order to allow readers to better understand, this article gives a more detailed description of Table 3 and the fracture to facilitate readers' understanding.

(After modification)In this paper, a laboratory hydraulic servo low-frequency fatigue testing machine is used to apply a load to simulate the actual service process of the sample. Follow the national standard GB/T 3075-2008 "Metal Material Fatigue Test Axial Force Control Method", using the lifting method to perform the fatigue test with 106 cycles for rolling 316L and laser selective additive manufacturing 316L,and fatigue data results Are shown in Table 3. According to the data in Table 3, the fatigue limits of rolling 316L and SLM 316L specimens are 370MPa and 200MPa, respectively, under the condition that no fracture occurs after 106 times. Without breaking 106 times, the stress level of SLM 316L only reached 54% of that of rolled 316L. As the stress level increases, the fatigue life of rolled 316L decreases rapidly. The fatigue life of laser-selected area additive manufacturing 316L decreases slowly with the increase of stress level. Sometimes with the increase of stress level, the fatigue life of laser-selected area additive manufacturing 316L Life expectancy will increase.

Figures 4a-c are the fracture morphologies of rolled 316 B5, and Figure 4a shows the macroscopic morphology of sample B5. It can be seen that there are fewer internal defects, obvious plastic deformation in the fracture, and obvious inclusion defects in the lower left corner of the fracture. The inclusion defects are located on the surface of the sample and spread radially around. Figure 4b shows an enlarged view of the fatigue source area of sample B5, which is also a further enlargement of Figure 4a. It can be seen that fatigue cracks originate from inclusion defects on the surface of the sample. After the crack is formed, it starts from the defect and spreads radially around. Defects under cyclic loading will cause the stress concentration to be higher than the surroundings, resulting in microcracks in the sample without a large external load. Fig. 4c is an enlarged view of the defect of the fracture surface of sample B5, which is also a further enlargement of Fig. 4b. It can be seen the existence of the tear ridge (inside the yellow ellipse), which is left when the cracks in the expansion process meet when they expand in all directions.

Table.3 Fatigue test data of rolled 316L and SLM-316L

Type

Sample No.

Maximum stress (Smax/MPa)

Stress amplitude (Sa/MPa)

Life cycle

Rolled

B-1

460

207

241870

Rolled

B-2

440

198

262350

Rolled

B-3

420

189

354397

Rolled

B-4

400

180

461752

Rolled

B-5

380

171

600691

Rolled

B-6

370

166.5

1000000

Rolled

B-7

360

162

1000000

SLM

S-1

300

135

421906

SLM

S-2

280

126

399303

SLM

S-3

260

117

465199

SLM

S-4

240

108

454712

SLM

S-5

220

99

530287

SLM

S-6

210

94.5

580493

SLM

S-7

200

90

1000000

Fig.4 Fracture morphology of rolling 316L and SLM-316L fatigue intiation: (a-c) rolling 316L, (d-h) SLM-316L.

Observe Figure 4d-f, Figure 4d-f is the fracture morphology of SLM-316L sample S6. Figure 4d shows the macro fracture morphology of sample S6. From the figure, it can be seen that there are defects such as sintered powder (in the yellow frame) and unfused pores (in the yellow circle) inside the sample. 4e-h is an enlarged view of the fracture morphology of sample S6, which is also a further enlargement of Figure 4d. As shown in Figure 4e, you can see the tearing ridges produced by the cracks on the left that meet the defects caused by unfusion. The cracks will expand further after encountering the defects during the propagation process. And as shown in Figure 4f, around the sintered powder, the cracks generated will expand to the surroundings. As shown in Fig. 4g, cracks are generated on the surface of the sample, and then expand forward, and tear ridges will also be generated when intersecting with other cracks. It can be found that the difference from rolled 316L is that, depending on the type of cracks, microcracks may occur on the surface of defects such as sintered powder, unfused, and pores. According to the location of the crack, the fatigue source of laser-selected area additive manufacturing 316L not only exists on the surface of the sample, but also exists inside the sample, and may also exist near the surface. Under the action of cyclic loading, stress concentration is prone to occur near the defect, especially on the surface of some sharp notches, the stress concentration factor is much higher than the surroundings. There are microcracks near defects such as sintered powder, LOF, pores, etc., indicating that these process defects promote the formation and propagation of cracks during the actual loading process of the sample. The materials used this time are added with the same parameters. There are defects in samples S1-S7, but some samples have many defects and some samples have few defects. As a result, the fa-tigue life of laser selective additive manufacturing of 316L is not obvious or the fatigue life is increased by the increase of stress level. If other parameters SLM-316L are selected, the internal defects of 316L may be reduced and the fatigue performance may be better, But the defect cannot be completely eliminated.Defects such as sintered powder, LOF, and pores are common defects in laser additive.

#The Materials journal require a specific format of references, authors must pay more attention in their writing. No reference is written according to the format required by the journal.

Response 8

The references in this article have been carefully revised according to the format of the journal.

#There are some grammar and typing mistakes.

Response 9

The grammar and spelling errors in the text were carefully revised.

#Please correct the typos.

Response 10

This article has carefully corrected typos.

#The authors must revise the entire manuscript.

Response 11

The author has carefully revised the full text.

Thank you for your suggestions on the article.